# SpaceWire-to-UWB Wireless Interface Units for Intra-spacecraft Communication Links

**DOI:** 10.3390/s23031363

**Published:** 2023-01-26

**Authors:** Rares-Calin Buta, Martin Drobczyk, Thomas Firchau, Andre Luebken, Tudor Petru Palade, Andra Pastrav, Emanuel Puschita

**Affiliations:** 1Communications Department, Technical University of Cluj-Napoca, 400114 Cluj-Napoca, Romania; 2Avionics Systems Department, German Aerospace Center, 28359 Bremen, Germany; 3National Institute for Research and Development of Isotopic and Molecular Technologies, 400293 Cluj-Napoca, Romania

**Keywords:** Eu: CROPIS mission, intra-spacecraft wireless communications, UWB transmissions

## Abstract

In the context of the Eu:CROPIS mission requirements, this paper aims to test and validate an intra-spacecraft wireless transmission carried between two SpW-to-UWB Wireless Interface Units (WIUs). The WIUs are designed to replace the on-board SpaceWire (SpW) connections of a spacecraft network. The novelty of this solution resides in prototyping and testing proprietary TRL6 WIUs for the implementation of both PDHU and CDHU units, which constitute a spacecraft network. The validation test scenarios employed in this paper were designed under the Eu:CROPIS mission system requirements as defined by the WiSAT-3 European Space Agency (ESA)-funded project. The SpW-to-UWB WIUs run a custom-built ISA100 over an IEEE 802.15.4 UWB PHY layer communication stack. The WIUs are evaluated based on four mission-specific performance test scenarios: (1) the link setup test, (2) the end-to-end delay test, (3) the maximum data rate test and (4) the housekeeping test. The validation test scenarios of the WIUs are carried out with the use of STAR-Dundee SpW-capable equipment. The test results demonstrate the reliability of the deployed SpW-to-UWB WIUs devices for UWB wireless communications carried out within a space shuttle. The SpW data were successfully transmitted across the intra-spacecraft wireless network in all experimental tests. The technology can be considered to be at the maturity level TRL6 (functionality demonstrated in relevant environment) for LEO missions.

## 1. Introduction

International space agencies (such as ESA, NASA, and JAXA) have taken numerous steps to investigate wireless technology and its applications in space projects. These agencies have shown an increasing interest in using wireless communications to replace, supplement, or extend wired data communications systems on-board satellites.

Even though wireless data transfer in space applications is challenging, a wireless solution could eliminate the expensive intra-spacecraft wiring process, make data communication harnesses easier to implement, increase the adaptability and flexibility of subsystem layouts, and provide additional space inside the satellite.

Spacecraft communications networks need shielding to reduce interferences and redundant links to enhance transmission dependability. By using the proper channel access strategies, a wireless solution might be in the position to offer the same amount of redundancy and lessen interference. This will lower, consequently, the overall launch cost and shorten the assembly, integration, and testing (AIT) period by decreasing the mass that needs to be launched into the orbit.

In the context of the Euglena and Combined Regenerative Organic Food Production in Space (Eu:CROPIS) satellite mission system requirements, this paper aims to test and validate a wireless transmission implemented within a spacecraft network comprising two data transfer units, namely the PDHU (payload data handling unit) and the CDHU (control data handling unit).

This manuscript extends our previous work presented in “A UWB solution for wireless intra-spacecraft transmissions of sensor and SpaceWire data” published by the *International Journal of Satellite Communications and Networking* [1]. Nonetheless, the key elements that outline the novelty of this paper and at the same time distinguish it from the previous work described in [1] derive from: (1) the use of validation test scenarios defined by Eu:CROPIS mission system requirements; (2) the use of proprietary, CDS-designed TRL6 WIU Engineering Qualification Models (EQM), which exhibit both a custom hardware design (compact solution for CDHU/PDHU implementation) and a custom software design (proprietary software running on an MCU); and (3) the use of SpW-capable equipment for WIU tests and validation. Through the three major elements mentioned above, the work presented in this manuscript displays advantages over the previous work described in [1,2,3].

WIUs’ implementation is based on the technical competences and results obtained during previous and ongoing collaborations between Control Data Systems (CDSs), the German Aerospace Center / Deutsches Zentrum für Luft- und Raumfahrt (DLR), and the Technical University of Cluj-Napoca / Universitatea Tehnică din Cluj-Napoca (TUC-N).

As such, the originality of this solution resides in prototyping and testing an SpW-to-UWB Wireless Interface Unit (WIU) for both the PDHU unit and the CDHU unit in the form of a WIU gateway and WIU node, respectively. The prototyped UWB wireless gateway and node run a custom-built ISA100 over an IEEE 802.15.4 UWB PHY layer communication stack.

The deployed SpW-to-UWB WIUs were evaluated based on four different performance test scenarios: (1) the link setup test, (2) the end-to-end delay test, (3) the maximum data rate test, and (4) the housekeeping test. The key performance indicators that were investigated and analyzed were the average end-to-end delay and the maximum achievable data rate. A STAR-Dundee SpW Brick Mk3 is used both as an SpW traffic generator and as an SpW traffic sink for (1) the data rate, (3) end-to-end latency, and (3) BER measurement. The SpW Brick Mk3 actually acts as a USB interface to SpW. It is suitable for use in SpW equipment evaluation, integration support, and system simulation.

The results show the reliability of the deployed solution for ultra-wideband (UWB) wireless communications carried out within a space shuttle between the PDHU and CDHU subsystems in a highly reflective propagation environment.

The key elements involved in this research are: (1) the use of Eu:CROPIS mission system requirements and validation test scenarios as defined by the WiSAT-3 European Space Agency (ESA)-funded project, (2) the use of TRL6 WIUs implemented by Control Data Systems (CDSs), and (3) the use of STAR-Dundee SpW-capable equipment (i.e., SpW Brick Mk2 and SpW Link Analyzer Mk2) available at the IntraSAT-Tech Centre at TUC-N for WIU tests and validation.

The remainder of this paper is organized as follows. In Section 2, available technologies and studies performed in the area of UWB communications are presented. In Section 3, the satellite mission context is detailed, and the actual system requirements are exhibited. In Section 4, the SpW-to-UWB WIU is described from the perspective of its hardware and software design and implementation. Next, in Section 5, the laboratory test scenarios are depicted, and in Section 6 the test results are addressed, and the performance of the implemented system is evaluated and analyzed. Finally, in Section 7, final discussions and conclusions complete the paper and the approached subject.

## 2. State of the Art of UWB (Ultra-Wideband) Communications

Even though wireless data transfer in space applications is difficult, a wireless solution could eliminate the expensive intra-spacecraft wiring process, make data communication harnesses easier to implement, increase the adaptability and flexibility of subsystem layouts, and save on space inside the satellite. Moreover, the wireless solution makes it possible to establish network redundancy in a more facile manner.

This will thus lower the overall launch cost and shorten the AIT period by lowering the mass that needs to be launched into orbit. In [4], a systematic study on how wireless communications can be adopted in intra-spacecraft systems is presented. It discusses the benefits of wireless over wired connections in terms of flexibility, adaptivity, redundancy realization, and interference suppression.

Due to its reputation [5,6] as a robust technology for multipath channel models, ultra-wideband (UWB) technology has been considered in most of the research conducted up to this point. The wired SpW [7] lines to be replaced, however, have a maximum nominal data rate of 200 Mbps and support high-throughput applications, whilst the UWB radio transceivers currently in use (such as the DecaWave DWM1000 transceiver) only offer a maximum nominal data rate of 27 Mbps [8].

The work described in [2] involves developing and testing a Wireless Sensor Network which connects intra-satellite payload sensors to a Payload Data Handling Unit (PDHU) using a specialized ISA100 industrial wireless communication protocol over an ultra-wideband RF communication channel. More specifically, the purpose of this work was to deploy a wireless payload network for intra-spacecraft communications, by adapting the ISA100 industrial protocol over the IEEE 802.15.4 PHY specifications. Three wirelessly enabled camera sensors and a wireless gateway were used as the test platform for all experiments. As test scenarios, telecommand messages for image acquisition from the PC application to each of the sensors were issued and, later on, the images were received by the PC application, thus showing the suitability of the ISA100 standard over UWB transmissions for intra-satellite communications.

The performance of the IEEE 802.15.4 standard in short-range indoor high-speed wireless communications has been studied in [9]. The IR-UWB physical layer proved to be a viable candidate to replace the wired spacecraft communications, as determined by analyzing through simulations the BER performances in multipath propagation and narrowband interference conditions.

In the context of classical wired intra-spacecraft communications, [3] presents a solution to replace, extend or complement the existing wired solutions for intra-satellite data transfer with wireless ones. In this sense, a UWB module called VN360 is presented. It implements a custom-built software stack of the ISA-100.11a protocol working over the IEEE 802.15.4 UWB PHY layer. The UWB solution was brought to TRL4, and it was tested in various environmental conditions, including a satellite mock-up. A reliable intra-spacecraft UWB connection could be established via the VN360 module.

The IntraSAT-Tech research center offered a UWB solution for wireless intra-satellite communications in the context of the Eu:CROPIS mission, which is presented and discussed in [3]. The primary goal was to replace the wired connections, such as SpW links or serial links, which were already present in the payload network and the spacecraft network segment, with wireless connectivity. In order to interface with the pre-existing on-board entities and conduct proper data transfer in the pre-defined spacecraft wireless network, particular sets of UWB nodes and gateways were prototyped. A special radio module was created to accomplish this duty, equipped with an appropriate UWB RF front end and a special communication stack. Sensor data were collected and sent from the payload network to the spacecraft network during the validation tests. The method was revealed to be suitable for the requirements of an intra-spacecraft network in terms of data integrity and delay.

A step forward realized by the IntraSAT-Tech research center led to the work described in [1,3], in which an SpW converter that converts SpW data into traffic adequate for an external interface (e.g., Ethernet, PCIe) and vice versa was developed. This approach overcomes the disadvantages of a wired intra-spacecraft network implementation, such as expensive wiring process, dry mass increase, reduced space, and communication infrastructure complexity. On the ZCU102 FPGA platform, the converter incorporates a UDP/IP-to-SpW converter and an SpW IP core [10]. The high-throughput SpW-to-Wireless bridge described in the current study uses this converter as a necessary first step. Additionally, a wireless converter (WIC) [11] was considered to provide the radio front end and wireless high-throughput connection for the upcoming SpW-to-WIC. The WIC was tested with a variety of antenna configurations using National Instruments devices.

Recent studies, such as [12,13], showed that wireless sensor networks are well suited for space applications when it comes to low-power applications, harness complexity reduction, and assembly, integration, and testing process (AIT time) simplification.

Focusing on wireless technologies in spacecraft applications, the German Aerospace Centre (DLR) proved that the IR-UWB transmission technique is suitable for stringent electromagnetic requirements [14], as the study in [5] shows that UWB technology overcomes the effects of a highly reflexive medium. The study considered a wireless experiment for localization and moving-object tracking deployed in the Columbus module of the International Space Station (ISS). One study [9] introduced inspaWSN, a novel wireless protocol convenient for intra-spacecraft communications. The protocol makes use of IR-UWB and an enhanced network protocol suitable for the requirements of an attitude and orbit control system (AOCS) implemented on the spacecraft networks. Another data network protocol built integrating the IR-UWB technology is presented in [15]. In [16], the DLR characterizes the performances of a COTS (commercial-off-the-shelf) DW1000 IR-UWB transceiver featured by Decawave under radiation conditions. The total ionizing dose effects and proton-induced effects are analyzed, and the results show that the device withstands LEO (low earth orbit) radiation conditions without any significant destructive effects.

The Wireless Compose-2 experiment for the International Space Station (ISS) is described in [17]. (WICO2). Considering low-power wireless applications in the medical and scientific fields, this experiment demonstrates the integration of a wireless network infrastructure in a commodity object. In this paper, a method for integrating wireless sensor networks (WSNs) within spacecraft using IR-UWB (impulse radio ultra-wideband) is presented. For the purpose of monitoring cardiovascular parameters in a microgravity setting, the WSN is integrated as part of a body area network. Additionally, ranging tests were carried out using IR-UWB technology. The power source for the sensor nodes was generated by harvesting energy from internal light sources.

Still in the technological range of IR-UWB, [18] presents an application that enables data transfer with low latencies and large data rates, reaching performances of up to 341 kbps. The MAC layer of impulse radio ultra-wideband (IR-UWB), which allows for larger data rates in comparison to the conventional unmodified MAC layer, was extended in a low latency and deterministic manner in order to permit the increase in the data rate. Thus, the IEEE 802.15.4 standard was extended to support higher data rates while being compliant with the MAC layer for UWB transmissions. The solution permits transferring not only low-latency sensor data, but also scientific data from the payload network.

## 3. System Requirements of the Satellite Mission

The verification scenario of the SpW-to-UWB system is based on the Eu:CROPIS satellite mission profile. The Eu:CROPIS mission is placed in a sun-synchronous, low earth orbit (LEO) with an approximate altitude of 600 km and is thus subjected to the typical environmental conditions of the LEO missions, especially regarding the radiation effects.

The test scenario for this system consists of the central command and data handling unit (CDHU) and an attached payload network, which forms the payload data handling unit (PDHU). SpW is an ESA standard now established as one of the main spacecraft on-board data-handling networks. Typically, these components are connected via a high-speed SpW link that is configured to a data rate of at least 40 Mbps. The SpW link is used to transfer CCSDS (Consultative Committee for Space Data System)-compliant packets between the CDHU and PDHU using the CCSDS space packet protocol.

The SpW protocol supports building complex networks with multiple endpoints connected with the help of specific routers. It also supports multiple protocols in parallel by using a protocol identifier in each SpW packet.

As such, the required functionality of the WIUs should be similar to that of an SpW router, as indicated in Figure 1.

Moreover, the WIUs should be able to support exchanging CCSDS space packet test data over UWB between CDHU and PDHU simulators, as presented in Figure 2. Between the two PDHU/CDHU simulators and the UWB WIUs, the SpW traffic is monitored by means of an SpW Analyzer (i.e., STAR-Dundee SpW Link Analyser), which captures and displays bi-directional SpW traffic traveling over the link.

For the WIU equipment to be compatible with future protocol extensions, it is recommended the implementation of its functionalities is kept compatible with the requirements indicated in the SpW standard white paper.

Besides the functional interface, the system also needs to implement an administration interface that is integrated into the SpW connection whose role is to minimize the number of physical interfaces employed in a real-world application. The administration interface utilizes a standardized interface protocol that is employed for other spacecraft PDHUs. This interface should allow the CDHU to read housekeeping data from the WIUs on both the UWB and the SpW interface.

To meet functional requirements, the interface units need to implement a routing function for the incoming SpW packets and must pass the incoming data via the wireless link unaltered to the other port of the bridge.

Non-functional requirements include product assurance for EEE parts, i.e., for COTS component utilization and derating, as well as for reliability, manufacturing, and electrical requirements mandated by the ECSS standards. Environmental requirements for mechanical loads, thermal design and limits, radiation-induced effects for total ionizing dose (TID) and single-event effects (SEE) were chosen based on the Eu:CROPIS mission profile and the typical environment that is expected there.

## 4. SpW-to-UWB WIU Design and Implementation

In the following sub-sections, the hardware design and software implementation of the WIU gateway is briefly presented.

### 4.1. WIU Gateway and Node Hardware Design

The main architectural blocks of the WIU are the following: (1) the SpW interface, used to connect to the S/C (spacecraft) host, (2) the UWB interface, employed to communicate over the UWB wireless network, (3) the MCU block, which comprises one CPU along with redundant RAM and flash chips, (4) the USB interface, used only for development and MAIT (manufacturing, assembly, integration, and test), and (5) the power block, used to power the WIU. 

Figure 3 shows the physical appearance of the SpW-to-UWB WIU hardware.

The enclosure of the WIU device is made of a cover that is 3D-printed from ABS+ material and mounted onto an aluminum base plate. The base plate is 2 mm thick and has three slotted holes on each side for convenient fastening of the WIU to the spacecraft structure.

Figure 4 shows the enclosure design and dimensions.

The whole construction is held together by four countersunk screws, which go through the aluminum base plate and four spacers soldered onto the PCBA in order to secure the top cover by screwing into the cover’s four mounting holes with threaded metallic insertions, as can be seen in Figure 4.

The enclosure employs 1 micro-D connector with 11 pins, 3 pins dedicated to power supply (1 pin for VCC, 1 pin for GND, and 1 pin left unused), and 8 pins dedicated to SpW connectivity (Din+, Din-. Sin+, Sin-, Dout+, Dout-, Sout+, Sout-).

The UWB antenna is fixed on the inside of the top cover into a specifically designed recessed slot using the included 3M adhesive tape [19].

### 4.2. WIU Gateway and Node Software Implementation

The wireless communication stack of the WIU device features a custom ISA100 protocol implementation over the IEEE 802.15.4 UWB PHY layer. This proprietary software stack solution is exhaustively described in [2,3] and it will be outlined in a few paragraphs, as follows.

UWB is a transmission technology based on short-impulse radio that is able to deal with propagation media affected by severe multipath environments that can be characterized as highly reflective environments, as is the case for spacecraft and aircraft [20]. In current UWB chipsets [21], 2ns and 1ns impulses are used for signal modulation, which result in 500 MHz and 1 GHz channel bandwidths, respectively. Consequently, a wider power spectral density distribution is obtained as compared to the narrow-band modulation techniques.

ISA100 [22] is an open standard employed mainly in wireless sensor networks (WSNs). It was originally created for the oil and gas industry. However, it was later adopted by the Consultative Committee for Space Data Systems (CCSDS) [23] as it was found suitable by NASA for the aerospace industry.

The ISA100 standard usually specifies the employment of IEEE 802.15.4 in 2.4 GHz for the PHY layer. In this work, ISA100 displays a custom PHY implementation over IEEE 802.15.4 by employing the UWB technology instead of the original IEEE 802.15.4 PHY in the 2.4 GHz band. This custom implementation permits higher data rates and the inherent benefit of using UWB in highly reflective media.

Regarding the medium access layer (MAC), the ISA100 over IEEE 802.15.4 PHY UWB employs a proprietary TDMA scheme in order to increase the data throughput [2]. A configurable access scheme integrated in the wireless nodes pre-allocates the time slots.

Next, regarding the network layer, it incorporates a 6lowPAN implementation, which is in charge of translating packets from IEEE 802.15.4 to IPv6 packets. Then, the UDP protocol is employed at the transport layer. 

The relationship between the implemented proprietary communication stack and the OSI model, together with the differences from the standard ISA100 specifications, is illustrated and described more in detail in [2].

## 5. Laboratory Test Scenarios Description

Next, the laboratory test scenarios are presented. The scenarios were defined within the WiSAT-3 ESA-funded project [19]. The laboratory tests were carried out both on CDS and TUC-N premises (in Romania) as preliminary tests, and at DLR facilities as final validation tests (in Germany).

The system evaluation was based on four basic steps: (1) verification of the test setup, (2) link test for the basic connectivity testing, (3) housekeeping data collection from the WIUs, and (4) performance measurements of the end-to-end SpW transmission.

The test scenario execution steps are illustrated in Figure 5.

### 5.1. Verification of Test Setup

Figure 6 displays the verified test setup, which consisted of the following components: (1) the power source, which supplied the proper voltage and current to the WIU node and gateway, (2) WIU node, (3) WIU gateway, (4) SpW Brick Mk3, (5) wireless packet sniffer and (6) laptop PC, which controls the SpW Brick Mk3 and the wireless packet sniffer.

### 5.2. Link Test

The next step was the first actual use of the test setup. In order to verify the basic setup configuration, every possible communication path between the Laptop PC and the WIUs was tested using a short link test request.

For example, the PC sends a link test request over the direct SpW connection to the administration interface of the WIU gateway. The gateway sends a link test response over the same direct connection. If the response is correctly received by the PC, the connection can be assumed to be functional. 

This process was repeated with the WIU node over the wireless link by employing the WIU gateway as a transparent relay for the link test request and link test response messages.

### 5.3. Housekeeping Data Collection

The housekeeping data collection test was used to verify additional functionalities of the administration interface of each WIU. The interface provides housekeeping data such as the status and configuration for both the physical SpW interface and the wireless interface.

Additionally, error counters can be accessed for each physical interface.

### 5.4. Performance Measurements

In order to provide a first assessment of the data transfer performance in the chosen SpW bridge scenario, additional performance characterization tests were performed after the functional tests. Since SpW traffic is routed transparently over the bridge, the standard STAR-Dundee utilities were used, with minor modifications for additional statistics output.

Performance was measured in terms of end-to-end packet latency and maximum data rate. In both cases, we gradually increased packet sizes in order to find any tipping or saturation points. Both the latency and maximum data rate test were first performed with a direct, wired connection between the ports of the SpW interface in order to establish a baseline.

#### 5.4.1. End-to-end Delay Test

The end-to-end delay test application involves the employment of the STAR-Dundee Brick Mk3 equipment, and the Performance Tester application also provided by STAR-Dundee. The precise synchronization between the SpW traffic generator and the SpW traffic sink is necessary for the average end-to-end delay measurements. Consequently, the STAR-Dundee Brick Mk3 is further linked in a loopback mode by connecting one port of the equipment to the emitting WIU and the second port to the receiving WIU. Thus, a local synchronization of the STAR-Dundee Brick Mk3 with itself is achieved.

Regarding the Performance Tester application, in its settings menu, it was configured to transmit packets of variable sizes, starting from 1 byte to 4096 bytes, with logarithmic increments. For each packet size, the STAR-Dundee Brick Mk3 device generated 125,000 packets. In this case, the latency measurement option was chosen from the application menu. To measure the average end-to-end latency, the software of the application is set to generate one packet of a specific size on one port, and to wait for that specific packet to be received on the other port. After the first packet is received, the application passes to the second packet to be transmitted, and the process is repeated until all 125,000 packets of a particular size are transmitted. Finally, the application computes the average of the recorded packet delays and stores the final value in a text file.

#### 5.4.2. Maximum Data Rate Test

The maximum data rate test involved, as in the previous case, the usage of the STAR-Dundee Brick Mk3 and its associated Performance Tester application. In the maximum data rate test, multiple packets are sent and received, and the time between the first packet that was transmitted and the last packet that was received is measured. The data rate, the packet rate, and the average packet time are then determined using packet sizes and the recorded timestamps. 

The application was set from its settings menu to measure the maximum data rate achievable by the system under the test involving connection between the two ports of the STAR-Dundee Brick Mk3 device. The STAR-Dundee equipment generates in the same manner 125,000 packets of variable sizes, starting from 1 byte to 4096 bytes, through a logarithmic increment. This time, the packets are transmitted not one by one, but in bursts of 1000 packets, thus measuring the timing between the first packet that was sent up and the last one that was received, in order to obtain the average data rate of the packet bursts.

## 6. Performance Evaluation and Test Results

### 6.1. Preliminary Evaluation and Results

The key performance indicators that were evaluated through the four aforementioned test scenarios during the preliminary test session carried out on the TUC-N and CDS premises (in Romania) are (1) the maximum data rate and (2) the average end-to-end delay. The test suite employed the CDS SpW-to-UWB WIU gateway and node, and STAR-Dundee instrumentation (STAR-Dundee featured SpW Brick Mk3). The correspondence between the packet size and the number of packets sent over radio is presented in Table 1.

Due to the data packet fragmentation for packets larger than 800 bytes, in the case of 1024-byte packets, 2048-byte packets and 4096-byte packets, the number of packets sent over radio was different. It can be obtained by dividing the packet size by 800 bytes and rounding up the result. The total number of packets generated during a latency test or maximum data rate test was 2,625,000 bytes, as can be computed from Table 1 by summing up all the elements from the second column.

#### 6.1.1. Latency Test Gateway-to-Node

Table 2, Table 3, Table 4 and Table 5 display the results obtained after the UWB packet transfer between the WIC UWB gateway (CDHU) and WIC UWB node (PDHU). The average end-to-end delay and the average maximum data rate, respectively, were investigated.

In the case of the latency test performed between the WIU gateway and WIU node, it can be noticed that the average end-to-end packet delay was 5112 ms with a standard deviation of 3.13 ms, as can be inferred from Table 2. When the transmitted packet size exceeds the value of 800 bytes, fragmentation intervenes, such that a packet of 1024 bytes will be fragmented into two sub-packets, a packet of 2048 bytes will be fragmented into 3 sub-packets and a packet of 4096 bytes will be fragmented further into 6 sub-packets. This is the reason why the average end-to-end delay of 1024-, 2048-, and 4096-byte packets was larger. Moreover, the larger the packet, the more time-consuming the error-checking procedure through which the data bytes of packets stored in the RAM memory are checked against their corresponding checksum headers. 

#### 6.1.2. Latency Test Node-to-Gateway

In the case of the latency test performed with packets transmitted from the WIU node to the WIU gateway, the average end-to-end packet delay was 5115 ms with a standard deviation of 7.5 ms, as can be deduced from Table 3. As in the previous case, packets with a size exceeding 800 bytes were fragmented (i.e., packets of 1024 bytes are fragmented into two sub-packets, packets of 2048 bytes will be fragmented into 3 sub-packets and packets of 4096 bytes will be fragmented further into 6 sub-packets), which led to an increase in the average end-to-end delay. 

On the other hand, this delay increase was caused by the memory-scrubbing procedure which checks the data bytes stored in the RAM memory of the WIU gateway/node against a checksum header. As a result, for 1024-byte packets, the average end-to-end delay was 10,217 ms, for 2048 bytes, the average end-to-end delay was 15,326 ms, and for packet sizes of 4096 bytes, the average end-to-end delay was 35,754 ms.

#### 6.1.3. Maximum Data Rate Gateway-to-Node

The maximum data rate test with packets generated by the gateway WIU and received by the node WIU can be seen in Table 4. It can be immediately noticed that the packet rate was constant around an average of 195.58 packets/s until the size of 512 bytes. As the packet size exceeded the value of 800 bytes, the software stack running on the WIU node and gateway performed packet fragmentation. The fragmentation caused a decrease in the packet rate from an average of 195.58 packets/s to 97.87 packets/s for packet sizes of 1024 bytes, 65.24 packets/s for a packet size of 2048 bytes, and 27.96 packets/s for a packet size of 4096 bytes.

On the other hand, an exponential increase in the recorded data rate can be noticed with the increase in packet size, such that the maximum data rate that was obtained measured 1.07 Mbps for the maximum of 2048-byte packet. The 150 kbps decrease from the data rate recorded for a packet size of 2048 bytes to the one recorded for a packet size of 4096 bytes was caused by the memory-scrubbing procedure, which consists of reading each RAM memory location of the WIU and correcting bit errors if there are any or raising a flag if two or more errors are identified.

#### 6.1.4. Maximum Data Rate Node-to-Gateway

Regarding maximum data rate measurements in the case of data transmitted from the WIU node to the WIU gateway, the results preserve their characteristics as in the previous scenario, as can be seen in Table 5. In light of this, the data rate was increasing concomitantly with the packet size, except for the 4096-byte packets, where a small decrease in the data rate can be noted. This data rate downturn, as it was specified several times earlier, is caused by the memory-scrubbing procedure, which becomes increasingly cumbersome as the packet size increases.

What is more, the packet rate remained constant around an average of 195.63 packet/s with a standard deviation of 0.2 packets/s. However, due to the memory scrubbing and packet fragmentation for packets of sizes over 800 bytes, the packet rate value changed accordingly. On this basis, for a packet size of 1024 bytes, the recorded packet rate was 94.15 packets/s, for a packet size of 2048 bytes, the recorded packet rate was 65.26 packets/s, and for a packet size of 4096 bytes, the recorded packet rate was 27.97 packets/s.

#### 6.1.5. Housekeeping Data Collection Debugging Results

Table 6, Table 7, Table 8 and Table 9 indicate the number of packets and bytes received on the radio and SpW interfaces, respectively, for both the WIC UWB gateway (CDHU) and the WIC UWB node (PDHU).

Test results were gathered from both the WIC UWB gateway (CDHU) and WIC UWB node (PDHU) after a housekeeping command was issued from the STAR-Dundee Brick Mk3 device.

Subsequently, from the figures recorded in Table 6, one can determine that the WIU gateway sent as many as 5,250,000 packets over the radio interface, namely 2,625,000 packets for the latency test and 2,625,000 packets for the maximum data rate test. The same WIU received on the radio interface 10,500,000 packets, corresponding to 5,250,000 packets received from the WIU node and the same number of acknowledgment packets.

On the other hand, from the figures recorded in Table 8, the WIU node also sent as many as 5,250,000 packets over the radio interface, namely 2,625,000 packets for the latency test and 2,625,000 packets for the maximum data rate test. Additionally, the number of packets received over the radio interface was 10,500,001. This was the same as in the case of the WIU gateway, plus an additional packet corresponding to the housekeeping command received by the WIU node via the WIU gateway over the radio interface.

Concerning the data displayed in Table 7 and Table 9, here are recorded the number of bytes transmitted and received by the WIU gateway and WIU node over the SpW interface. An important aspect must be noted here. The length of a packet received by a WIU device on the SpW interface is 2 bytes larger than the length of a packet transmitted by the other WIU device on the SpW interface. Consequently, for a latency test or maximum data rate test, the number of bytes received on the SpW interface by one of the WIU devices is 3,250,000 bytes larger than the number of bytes sent on the SpW interface by the other WIU device. This must be considered, as those two specific bytes contain the routing information, and they are discarded as soon as the packet is received by the WIU device, and as the routing information is read.

As such, the WIU gateway received 239,962,500 bytes over the SpW interface, from which 38 bytes were from the housekeeping request. In contrast, the WIU node sent 2,334,625,000 bytes over the SpW interface (i.e., 239,962,500 bytes − 3,250,000 × 2 bytes = 2,334,625,000 bytes). On the other hand, the WIU node received 2,054,250,000 bytes over the SpW interface, whereas the WIU gateway transmitted 2,047,752,510 bytes over SpW (i.e., 2,054,250,000 bytes − 3,250,000 × 2 bytes plus 2510 erroneous bytes).

### 6.2. Final Evaluation and Test Results

The key performance indicators evaluated in the final tests performed at DLR facilities (in Germany) are (1) the maximum data rate and (2) the end-to-end delay.

After completing the functional testing, additional performance characterization tests were conducted in order to offer a preliminary evaluation of the data transfer performance in the chosen SpW bridge scenario.

The maximum data rate and end-to-end packet latency were used for system performance measurement. Since SpW communication is transparently forwarded over the bridge, only minimal adjustments were made to the normal STAR-Dundee utilities to provide more statistics.

In order to create a baseline, the SpW interface’s ports were first linked together for the latency and maximum data rate tests.

Figure 7 shows a comparison of maximum data rate over packet size for direct, wired connection and in both directions over the wireless bridge. The latter is nearly identical and therefore not discernable in the graph. H stands for host, WIU 1 stands for WIU gateway, WIU 2 stands for WIU node. Three data transfer scenarios were evaluated: (1) wireless data transmitted from host to WIU Node to WIU Gateway and back to host, (2) wireless data transmitted from host to WIU Gateway to WIU Node and back to host, and (3) wired data transmitted from host to host.

The wired connection has a physical link speed of 100 Mbit/s. At packet sizes over 1000 bytes, the link becomes saturated at a maximum net data rate of about 80 Mbit/s and delays in the host system no longer dominate the maximum data rate.

The wireless link is configured to a physical link speed of 6.8 Mbit/s. A similar saturation occurs at packet sizes over 512 bytes with a maximum net data rate of 1 Mbit/s.

For a direct connection, the latency, as shown in Figure 8, is almost constant for package sizes from 2 to 4096 bytes. The large deviation for small packet sizes of 1 byte can be explained by the high frequency of function calls that make interruptions visible to the host operating system.

The same effect can be observed in the latency measurements for data transmission over the wireless bridge. Dedicated measurements were taken for both directions of the bridge, which are shown in Figure 9 and Figure 10.

As expected, additional stops in the transmission path added latency. While a direct connection had an average latency of around 2 ms, the employment of the wireless nodes as bridges resulted in an average latency of around 7 ms. For packet sizes over 100 bytes, the latency started to rise. In order to minimize latency, a custom ISA100 frame with a payload size of 800 bytes was used instead of the default ISA100 127-byte frame.

## 7. Discussion and Conclusions

SpW is a widely employed intra-spacecraft wired communications protocol that is subject to several drawbacks, such as cable harness complexity, increased spacecraft mass, and increased assembly, integration, and testing time, respectively. In the framework of an increased interest in the domain of wireless intra-spacecraft communications, this research article aimed at testing and validating a wireless transmission implemented within a spacecraft network consisting of two data transfer units, namely the PDHU (payload data handling unit) and the CDHU (control data handling unit).

Due to its reputation as a robust technology for multipath channel models, ultra-wideband (UWB) technology has been considered in most of the research conducted up to this point a suitable solution to replace, extend or complement the existing wired solutions for intra-satellite data transfer with wireless ones.

In the context of Eu:CROPIS mission requirements, the wired intra-spacecraft communication within a space shuttle is replaced by a two-segment wireless network comprising two architectural units, named PDHU (payload data handling unit) and CDHU (control data handling unit).

The aim of this paper was to test and validate a wireless transmission implemented within a spacecraft network comprising two data transfer units, namely the PDHU and the CDHU. The two data transfer units were implemented in the form of two SpW-to-UWB Wireless Interface Units (WIU) which have similar functionalities to an SpW router, and which are compatible with the requirements indicated in the SpW standard. The wireless communication stack of the WIU device is based on the ISA100 standard over IEEE 802.15.4 UWB PHY.

The work comprised in this research article is innovative in that it implements both the PDHU and CDHU units, which compose the spacecraft network (SN), by prototyping and testing proprietary TRL6 WIUs designed by the CDS company in Cluj-Napoca. The validation test scenarios utilized in this study were developed in accordance with the WiSAT-3 European Space Agency (ESA)-funded project’s Eu:CROPIS mission system criteria.

The system was evaluated based on four different performance test scenarios consisting of a link setup test, end-to-end delay test, maximum data rate test, and housekeeping test. The key performance indicators that were investigated and analyzed were the average end-to-end delay and the maximum data rate achievable. A STAR-Dundee Brick Mk3 device was used both as an SpW traffic generator and as an SpW traffic sink for the data rate and end-to-end latency measurement. An SpW Link Analyser Mk2 device was employed to monitor unobtrusively the traffic on the SpW link.

Regarding the end-to-end latency tests performed between the WIU gateway and WIU node, it could be noticed that the average end-to-end packet delay was around 7 ms as compared to the 2 ms delay in case of a direct connection between the CDHU and PDHU. Regarding the maximum data rate achieved in the case of gateway-to-node transmission, a data rate of 1 Mbps was measured for packet sizes over 512 bytes as compared to a wired connection with a physical link speed of 100 Mbps. This value was preserved for the node-to-gateway data transmission flow, as well.

The solution was validated for real traffic that was generated by on-board SpW traffic emulation devices, setting the technical readiness level (TRL) to the functional verification level in a laboratory environment (TRL4). Moreover, it must be stated that the WIU devices have been tested and validated in a relevant environment for space applications. As such, the WIU was subjected to a mechanical/vibration test, EMC test, TVAC test and radiation tests, which were finalized successfully, thus bringing the SpW-to-UWB WIU bridge to TRL6.

The results show the reliability of the proposed solution for ultra-wideband (UWB) wireless communications carried out within a space shuttle between the PDHU and CDHU subsystems in a highly reflective propagation environment.

To conclude, the WIUs did successfully perform all functional tests as defined in accordance with the WiSAT-3, European Space Agency (ESA)-funded project, namely the Eu:CROPIS mission system requirements. The SpW data were successfully transmitted across the intra-spacecraft wireless network in all experimental tests. As was expected, the adopted architecture proved to be viable and functional. The technology can be considered to be at the maturity level TRL6 (functionality demonstrated in relevant environment) for LEO missions.

The suggested approach can be taken into consideration for future satellite missions as well, even though it is based on the needs of the Eu:CROPIS mission. SpW, the top communication system for both present and future spacecraft, was utilized in this approach. The solution uses ISA100 and IEEE 802.15.4 UWB PHY, which have been shown to be the most dependable wireless intra-satellite communication technologies among those already studied by the CCSDS [5]. 

## Figures and Tables

**Figure 1 sensors-23-01363-f001:**
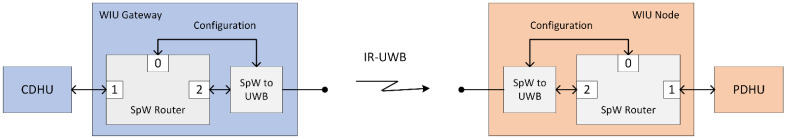
System architecture: wired-to-wireless requirements (the numbers indicated on the SpW Router (i.e., 0, 1, and 2) represent the port number of the SpW Routers).

**Figure 2 sensors-23-01363-f002:**
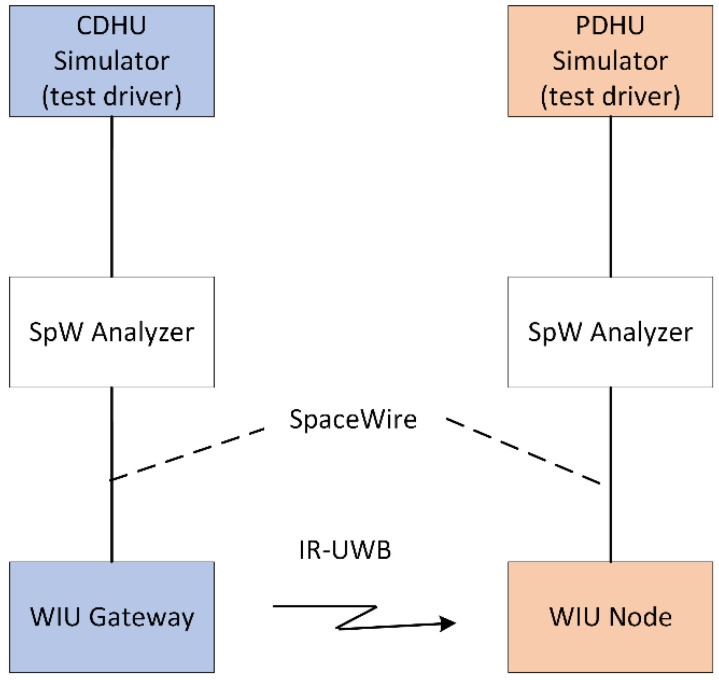
System architecture: wired-to-wireless requirements.

**Figure 3 sensors-23-01363-f003:**
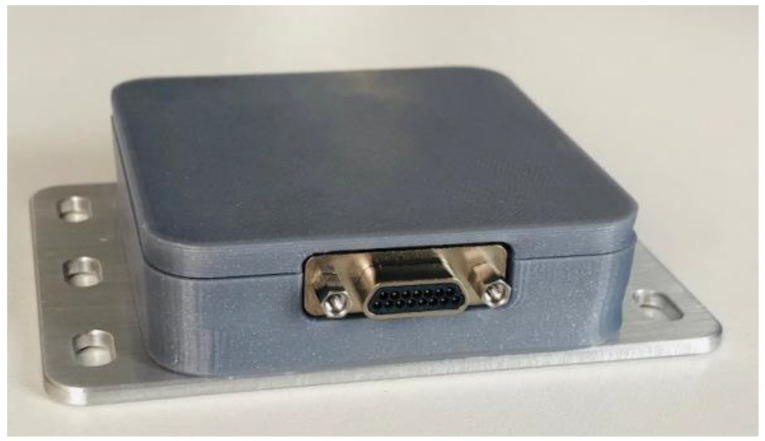
SpW-to-UWB WIU hardware.

**Figure 4 sensors-23-01363-f004:**
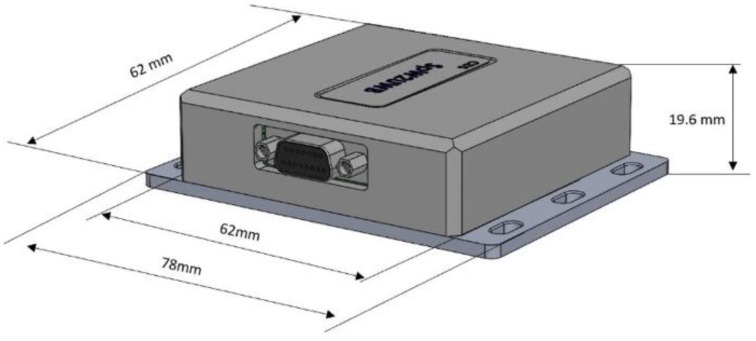
Enclosure dimensions.

**Figure 5 sensors-23-01363-f005:**
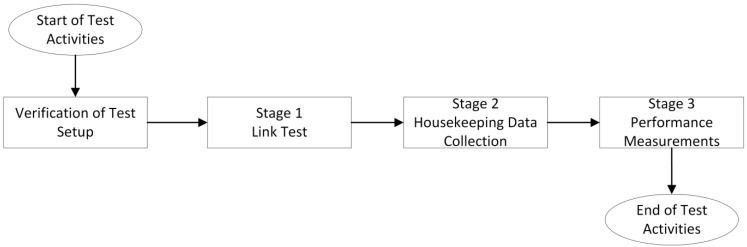
Test scenario execution steps.

**Figure 6 sensors-23-01363-f006:**
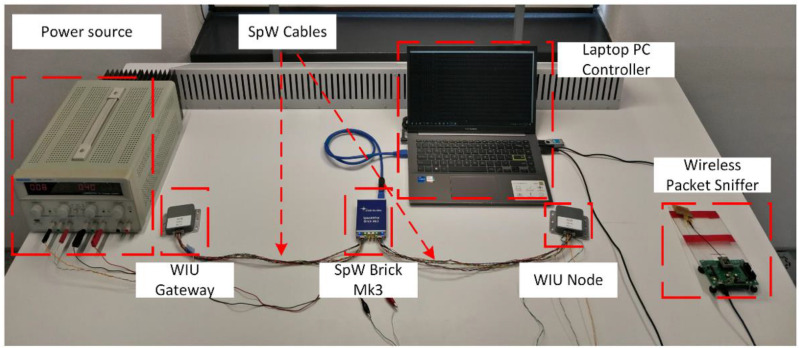
Testbed implementation (on Control Data Systems premises in Cluj-Napoca, Romania).

**Figure 7 sensors-23-01363-f007:**
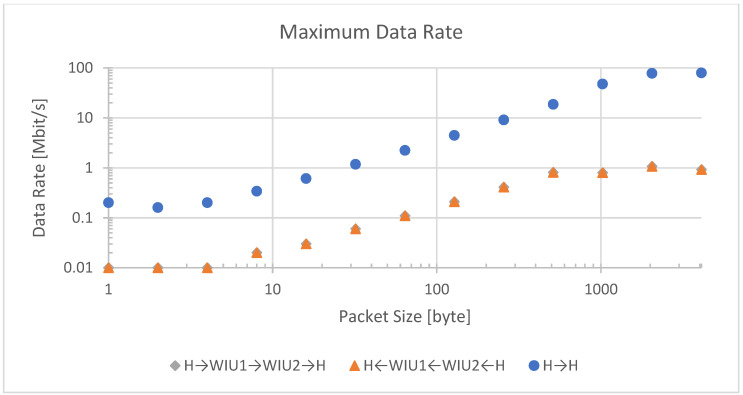
Maximum data rate for packages from 1 to 4096 bytes.

**Figure 8 sensors-23-01363-f008:**
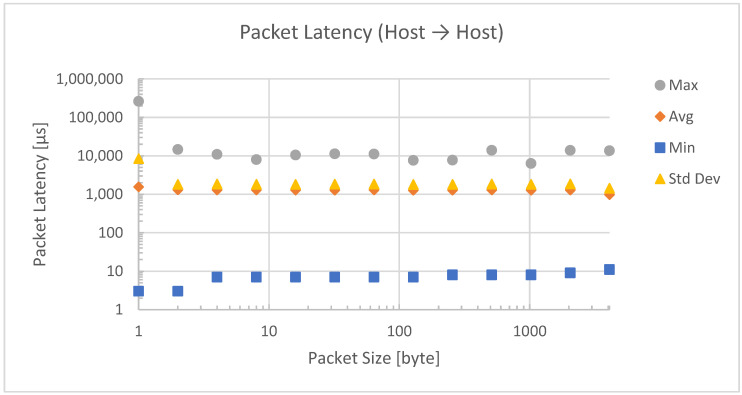
Packet latency for a direct host-to-host connection.

**Figure 9 sensors-23-01363-f009:**
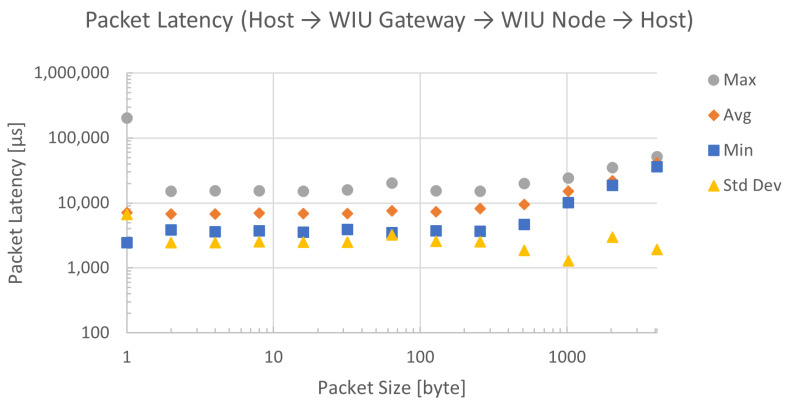
Packet latency for a bridged connection over WIU gateway to WIU node.

**Figure 10 sensors-23-01363-f010:**
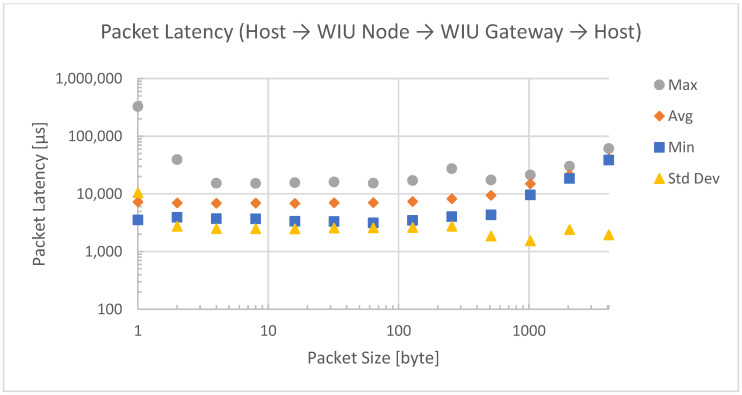
Packet latency for a bridged connection over WIU node to WIU gateway.

**Table 1 sensors-23-01363-t001:** WIC UWB GW (CDHU) to WIC UWB node (PDHU) average end-to-end delay.

Packet Size (bytes)	Number of Packets Sent Over Radio
1	125,000
2	125,000
4	125,000
8	125,000
16	125,000
32	125,000
64	125,000
128	125,000
256	125,000
512	125,000
1024	250,000
2048	375,000
4096	750,000

**Table 2 sensors-23-01363-t002:** WIC UWB GW (CDHU) to WIC UWB node (PDHU) average end-to-end delay.

Packet Size (Bytes)	Average Packet Time (ms)
1	5117.82
2	5115.25
4	5113.12
8	5114.24
16	5109.7
32	5110.18
64	5109.65
128	5109.62
256	5109
512	5109.18
1024	10,218.11
2048	20,355.54
4096	38,256.45

**Table 3 sensors-23-01363-t003:** WIC UWB node (PDHU) to WIC UWB GW (CDHU) average end-to-end delay.

Packet Size (bytes)	Average Packet Time (ms)
1	5129.82
2	5119.5
4	5114.82
8	5120.9
16	5110.16
32	5109.96
64	5109.09
128	5108.89
256	5109.1
512	5109.47
1024	10,217.03
2048	15,326.47
4096	35,754.62

**Table 4 sensors-23-01363-t004:** WIC UWB GW (CDHU) to WIC UWB node (PDHU) maximum data rate.

Packet Size (Bytes)	Data Rate (Mbps)	Packet Rate (Packets/s)
1	0	195.42
2	0.01	195.39
4	0.01	195.56
8	0.02	195.5
16	0.03	195.71
32	0.05	195.7
64	0.1	195.71
128	0.2	195.71
256	0.4	195.73
512	0.8	195.71
1024	0.8	97.87
2048	1.07	65.24
4096	0.92	27.96

**Table 5 sensors-23-01363-t005:** WIC UWB node (PDHU) to WIC UWB GW (CDHU) maximum data rate.

Packet Size [Bytes]	Data Rate [Mbps]	Packet Rate [Packets/s]
1	0	195.19
2	0.01	195.55
4	0.01	195.65
8	0.02	195.55
16	0.03	195.72
32	0.05	195.75
64	0.1	195.81
128	0.2	195.82
256	0.4	195.82
512	0.81	195.78
1024	0.77	94.15
2048	1.07	65.26
4096	0.92	27.97

**Table 6 sensors-23-01363-t006:** Received UWB Housekeeping Data from WIU Gateway.

WIU 1 UWB HK:
TX Counter: 5,250,000
RX Counter: 10,500,000
TX Error Counter: 0
RX Error Counter: 2176
Channel: 5
Prf: 2

**Table 7 sensors-23-01363-t007:** Received SpW Housekeeping Data from WIU Gateway.

WIU 1 SpW HK:
Control Register: 0xe7040026
Status Register: 0x00a00000
Address Register: 0x00fc
TX Counter: 2,047,752,510
RX Counter: 2,341,875,038
TX Error Counter: 0

**Table 8 sensors-23-01363-t008:** Received UWB Housekeeping Data from WIU Node via WIU Gateway.

WIU 2 UWB HK:
TX Counter: 5,250,000
RX Counter: 10,500,001
TX Error Counter: 0
RX Error Counter: 2928
Channel: 5
Prf: 2

**Table 9 sensors-23-01363-t009:** Received SpW Housekeeping Data from WIU Node via WIU Gateway.

WIU 2 SpW HK:
Control Register: 0xe7040026
Status Register: 0x00a00000
Address Register: 0x00fc
TX Counter: 239,962,500
RX Counter: 2,054,250,000
TX Error Counter: 0

## Data Availability

Not applicable

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
