# Peer review of "SpaceWire-to-UWB Wireless Interface Units for Intra-spacecraft Communication Links"

_sensors, 2023, doi:10.3390/s23031363_

Round 1

Reviewer 1 Report

Whilst the English is generally quite good, there are quite a few minor grammatical errors, and a careful read thoroughly.

Author Response

We thank the reviewers for their constructive comments and the insightful feedback. We are grateful for the opportunity to revise our paper. We addressed all the corrections and issues raised by the three reviewers and added the needed clarifications. Below is our detailed reply to all the comments and suggestions.

Authors Response to Reviewer #1 Comments

[Reviewer #1 Comments and suggestions]: Whilst the English is generally quite good, there are quite a few minor grammatical errors, and a careful read thoroughly.

[Reply]: Thank you for your suggestion. Indeed, after a more careful reading of the text, some grammatical mistakes have been discovered and corrected. We hope that no other mistakes have slipped our attention.

Also, in the following, the lines highlighted in yellow represent the text in the article which was modified according to the reviewer’s comments.

The following grammatical errors have been identified and corrected:

  1. Added a “.” in section Introduction in Finally, in Section 7 final discussions and conclusions complete the paper and the approached subject. (see line 104)
  2. Replaced “good” with “viable” in State of the art of UWB (ultra-wideband) communications in The IR-UWB physical layer proved to be a viable candidate to replace the wired spacecraft communications (see line 135)
  3. Replaced “that” with “which” in section State of the art of UWB (ultra-wideband) communications in … which were already present in the payload network (see line 148)
  4. Replaced “an highly…” with “a highly…” in section State of the art of UWB (ultra-wideband) communications in …as the study in [3] shows that UWB technology overcomes the effects of a highly… (see line 174)
  5. Replaced “with regard to” with “regarding” in section System requirements of the satellite mission in …especially regarding the radiation effects. (see line 206)
  6. Replaced “utilization” with “help” and replaced “connected together” with “connected” in section System requirements of the satellite mission in SpW protocol supports building complex networks with multiple endpoints connected with the help of specific routers. (see line 216)
  7. Deleted “very” from “very similar to” in section System requirements of the satellite mission in the required functionality of the WIUs should be similar to that of a SpW router (see line 218)
  8. Replaced “In order for” with “For” in section System requirements of the satellite mission in For the WIU equipment to be compatible (see line 229)
  9. Replaced “UWB is a transmission” with “UWB is transmission” in section 2. WIU Gateway and Node software implementation in UWB is a transmission… (see line 281)
  10. Replaced “with the exception for” with “except for” in section 1.4 Maximum data rate Node-to-Gateway in …the data rate is increasing concomitantly with the packet size increase, except for the 4096 bytes packets where a small decrease… (see line 451)
  11. Replaced “more and more” with “increasingly” in section 1.4 Maximum data rate Node-to-Gateway in … is caused by the memory scrubbing procedure which becomes increasingly cumbersome… (see line 454)
  12. Deleted “directly” in section 6.2 Final evaluation and test results in …the SpW interface's ports were first linked together for the latency and maximum data rate tests… (see line 511)

Reviewer 2 Report

The authors implement the SpW-to-UWBs WIUs test and evaluate the delay and data test rate in many different scenarios such as increasing packet rate and packet sizes. The results show that the proposed implementation is validated in the communication for the space applications.

Please check the typos. Please find more comments in the attached file.

Author Response

We thank the reviewers for their constructive comments and the insightful feedback. We are grateful for the opportunity to revise our paper. We addressed all the corrections and issues raised by the three reviewers and added the needed clarifications. Below is our detailed reply to all the comments and suggestions.

Authors Response to Reviewer #2 Comments

[Reviewer #2 Comments and suggestions]: The authors implement the SpW-to-UWBs WIUs test and evaluate the delay and data test rate in many different scenarios such as increasing packet rate and packet sizes. The results show that the proposed implementation is validated in the communication for the space applications.

[Reply]: Thank you for your supportive comment and valuable feedback.

[Comments and suggestions for Authors]: Please check the typos. Please find more comments in the attached file.

[Reply]: Indeed, after a more careful reading of the text, some typos have been identified. We hope that no other mistakes have slipped our attention.

Also, in the following, the lines highlighted in yellow represent the text in the article which was modified according to the reviewer’s comments.

The following typos have been identified:

  1. Added a space before “*” in the authors list Emanuel Puschita 1, 3, * (see line 5)
  2. Corrected from “(1)” to “(2)” in the enumeration of the key elements involved in this research at Introduction. (see line 93)
  3. Deleted “is” before “overcomes” in section State of the art of UWB (ultra-wideband) communications in …as the study in [3] shows that UWB technology overcomes the effects of a highly… (see line 173)
  4. Added “,” after “i.e.” in section System requirements of the satellite mission in …include the product assurance for EEE parts, i.e., for COTS component… (see line 241)
  5. Replaced “UWB is a transmission” with “UWB is transmission” in section 2. WIU Gateway and Node software implementation in UWB is a transmission… (see line 281)
  6. Replaced “air-crafts” with “aircrafts” in section 2. WIU Gateway and Node software implementation in …as it the case of spacecrafts and aircrafts which… (see line 282)
  7. Replaced “with the exception for” with “except for” in section 1.4 Maximum data rate Node-to-Gateway in …the data rate is increasing concomitantly with the packet size increase, except for the 4096 bytes packets where a small decrease… (see line 451)
  8. Deleted a space at the beginning of the sentence and added a “.” at the end of it in section 2 Final evaluation and test results In order to minimize latency, a custom ISA100 frame with a payload size of 800 bytes was used instead of the default ISA100 127 bytes frame. (see line 541)

[Comment #1] 1. Do you consider the topic original or relevant in the field? Does it address a specific gap in the field?

The wireless communications for intra-spacecraft require high network reliability, flexible network topology, low complexity in assembly, integration, and testing. This research performs an experiment to deploy the UWB wireless network with current wireless standard IEEE 802.15.4 UWB PHY layer. This work is an extension of another work on reference [1].

Reference [1]: E. Puschita et al., “A UWB solution for wireless intra‐spacecraft transmissions of sensor and SpaceWire data,” Int. J. Satell. Commun. Netw., vol. 38, no. 1, pp. 41–61, Jan. 2020, doi: 10.1002/sat.1307.

[Reply]:  Both articles present an intra-spacecraft UWB wireless solution that meets the system requirements of the Euglena and Combined Regenerative Organic Food Production in Space (Eu:CROPIS) mission. However, the two papers differ in terms of: (1) Architecture/Network Topology, (2) Hardware & Software, and (3) Test and Validation scenarios.

Considering the Eu:CROPIS system requirements, the wired intra-spacecraft communication system is replaced with a two-segment wireless network comprising the spacecraft network (SN) which replaces the SpaceWire (SpW) link between the Command and Data Handling Unit (CDHU) and the Payload Data Handling Unit (PDHU), and the payload network (PN), which replaces the serial links between the PDHU and the scientific sensors on board the spacecraft.

As such, both solutions are based on the technical expertise and results acquired during previous and on-going collaborations between Control Data Systems (CDS), the German Aerospace Center / Deutsches Zentrum für Luft- und Raumfahrt (DLR), and the Technical University of Cluj-Napoca (TUC-N). However, there are several differences must be emphasized:

  1. There is a difference in the architecture/network topology presented in the two papers. On the one hand, in the current paper the scope is to test and validate a wireless transmission implemented within a SN composed of the PDHU and the CDHU units. On the other hand, the scope of paper [1] covers both the SN and the PN. Nevertheless, the accent was put on the PN by prototyping a set of UWB wireless gateways and nodes to handle the sensor data on the links between the PDHU and the PN UWB sensors. Nevertheless, both papers follow the architecture requirements specified by the Eu:CROPIS mission.
  2. There is a difference in the employed hardware and software. On the one hand, in the current paper, the originality resides in prototyping and testing a custom and compact SpW-to-UWB Wireless Interface Unit (WIU) for both the PDHU unit and the CDHU unit under the form of a WIU Gateway (TRL6) and WIU Node (TRL6), respectively. Moreover, the wireless communication stack of the WIU device is a proprietary software that runs on an MCU. On the other hand, in paper [1], a set of UWB wireless gateways and nodes were prototyped for the PN and SN (TRL 4). While the designed PN modules are compact prototypes, the SN modules make use of additional non-compact, commercial off-the-shelf (COTS) components. The SN was implemented only to facilitate an end-to-end transmission on the link between the CDHU and the PN UWB sensors (TRL 4). Additionally, the SN UWB Gateway and Node prototypes use an FPGA board running a SpW IP.
  3. There is a difference in the test and validation scenarios. On the one hand, the deployed WIUs are evaluated based on Eu:CROPIS mission requirements. The tests and validation procedures were defined by the Eu:CROPIS mission. The accent in this paper falls on testing the deployed WIUs according to the testing procedure defined by this specific mission, with the help of SpW traffic emulators. On the other hand, in paper [1], the testing and validation process wanted to demonstrate the functionality of a SpW bridge designed after the Eu:CROPIS mission topology requirements.

To conclude, the current work is a distinct research compared with the one referenced in [1] due to several reasons: (1) the designed architectures differ, as in the current paper there is a wireless transmission implemented within the SN of an intra-spacecraft network, whereas in [1] the emphasis is on the PN network implementation, (2) the hardware and software implementations of the designed modules are different in the two papers in discussion, and (3) the test and validation scenarios differ in the two papers in the sense that in the current paper the emphasis falls on the testing and validation procedures defined by the Eu:CROPIS mission, whereas in [1] only functional tests are considered.

The following details were included in the paper in section 1. Introduction (see lines 54-65):

This manuscript extends our previous work “A UWB solution for wireless intra‐spacecraft transmissions of sensor and SpaceWire data” published by the International Journal of Satellite Communications and Networking [1]. Nonetheless, the key elements that outline the novelty of this paper and in the same time distinguish it from the previous work described in [1] derive from: (1) the use of validation test scenarios defined by of Eu:CROPIS mission system requirements, (2) the use of proprietary, CDS designed, TRL6 WIU Engineering Qualification Models (EQM), which exhibit both a custom hardware design (compact solution for CDHU/PDHU implementation)  and custom software design (proprietary software running on an MCU), and (3) the use of SpW-capable equipment for WIUs tests and validation. Through the three major elements mentioned above, the work presented in this manuscript displays its benefits as compared to the previous work described in [1], [7], and [9].

[Comment #4] 4. What specific improvements should the authors consider regarding the methodology?

What further controls should be considered? The testbeds are used to perform the real experiment. In the future, the network parameters can be varied according to different MAC protocols which was not mentioned in this research.

[Reply] The medium access layer (MAC) protocol implementation of the ISA100 over IEEE 802.15.4 PHY UWB employs a custom TDMA scheme. A paragraph concerning this aspect has been included in the paper.

The following details were added in section 4.2. WIU Gateway and Node software implementation (see lines 297-299):

Regarding the medium access layer (MAC), the ISA100 over IEEE 802.15.4 PHY UWB employs a custom TDMA scheme. A configurable access scheme integrated in the wireless nodes pre-allocates the time-slot.

[Comment #7] 7. Pls include any additional comments on the tables and figures.

In figure 8,9,10: the label of figures should be consistent with the figure caption, for example: it is denoted as while it is denoted as the figure.

[Reply] The labels of the Figures 8, 9, 10 are now consistent with the figure captions as the vertical axis title has been modified from “Packet Time” to match the figure caption i.e., “Packet Latency”.

Reviewer 3 Report

The paper needs some clarifications:

-Clear the novelty of the manuscript in abstract and conclusion.

-Clear the benefits compared to previous works.

-Clear the experimental results in the abstract and conclusion.

-Improve the quality of Fig 5.

Author Response

We thank the reviewers for their constructive comments and the insightful feedback. We are grateful for the opportunity to revise our paper. We addressed all the corrections and issues raised by the three reviewers and added the needed clarifications. Below is our detailed reply to all the comments and suggestions.

Authors Response to Reviewer #3 Comments

[Reviewer #3 Comments and suggestions]: The paper needs some clarifications:

[Comment #1] Clear the novelty of the manuscript in abstract and conclusion.

[Reply]:  As it was stated in section 1. Introduction, the novelty of the work presented in this manuscript resides in prototyping and testing a SpW-to-UWB Wireless Interface Unit (WIU) for both the PDHU unit and the CDHU unit under the form of a WIU Gateway, and WIU Node, respectively. A key element of the work presented in this paper is the use of Eu:CROPIS mission system requirements and validation test scenarios as defined by the WiSAT-3 European Space Agency (ESA) funded project The prototyped UWB wireless gateway and node run a custom-built ISA100 over IEEE 802.15.4 UWB PHY layer communication stack.

Also, in the following, the lines highlighted in yellow represent the text in the article which was modified according to the reviewer’s comments.

For a better understanding of these aspects the Abstract was reconsidered (see lines 14-21):

In the context of Eu:CROPIS mission requirements, the scope of this paper is to test and validate an intra-spacecraft wireless transmission carried between two SpW-to-UWB Wireless Interface Units (WIUs). The WIUs are designed to replace the on-board SpaceWire (SpW) connections of a space-craft network. The novelty of this solution resides in prototyping and testing proprietary TRL6 WIUs for the implementation of both PDHU and CDHU units which constitute a spacecraft network. The validation test scenarios employed in this paper were designed under the Eu:CROPIS mission system requirements as defined by the WiSAT-3 European Space Agency (ESA) funded project.

Further on, the following lines will be inserted into the section 7. Discussion and conclusions (see lines 571-575 and lines 581-582):

The work comprised in this research article is innovative in that it implements both the PDHU and CDHU units, which compose the spacecraft network (SN), by prototyping and testing proprietary TRL6 WIUs designed by CDS company in Cluj-Napoca. The validation test scenarios utilized in this study were developed in accordance with the WiSAT-3 European Space Agency (ESA)-funded project's Eu:CROPIS mission system criteria.

A SpW Link Analyser Mk2 device was employed to monitor unobtrusively the traffic on the SpW link.

[Comment #2]: Clear the benefits compared to previous works.

[Reply]:  In the context of the wireless intra-spacecraft communications, a UWB-based solution is presented in [9]. It is a UWB module called VN360. Its purpose is to replace the wired communications within a spacecraft. It implements a custom-built software stack of the ISA-100.11a protocol working over IEEE 802.15.4 UWB PHY layer. The UWB solution was brought to TRL4, and it was tested in various environmental conditions, including a satellite mock-up. A reliable intra-spacecraft UWB connection could be established via the VN360 module.

The work described in [7] involves developing and testing a Wireless Sensor Network which connects intra-satellite payload sensors to a Payload Data Handling Unit (PDHU) through a custom software stack of the ISA-100.11a protocol over the IEEE 802.15.4 UWB PHY layer. Three wirelessly enabled camera sensors and a wireless Gateway were used as the test platform for all experiments.

In [1], the system requirements for the architecture of the wireless intra-satellite network are provided in the framework of the Eu:CROPIS mission. The wired intra-spacecraft communication system is replaced with a two-segment wireless network, comprising two architectural units, named Spacecraft Network (SN) and Payload Network (PN). As such, distinct sets of UWB gateways and nodes are prototyped to interface with the on-board entities and properly handle the data transmission in the resulting SN and PN units. The scope of paper [1] covered mainly the PN unit by prototyping a set of UWB wireless gateways and nodes to handle the sensor data on the links between the PDHU and the PN UWB sensors.

The key elements that outline the benefits of the current paper as compared to the previous work, derive from: (1) the use of validation test scenarios defined by of Eu:CROPIS mission system requirements, (2) the use of proprietary, CDS designed, WIU Engineering Qualification Models (EQM), which exhibit both a custom hardware design (compact solution for CDHU/PDHU implementation) and custom software design (proprietary software running on an MCU), and (3) the use of SpW-capable equipment for WIUs tests and validation process.

On the above issue, the following details were included in the paper in section 1. Introduction (see lines 54-65):

This manuscript extends our previous work “A UWB solution for wireless intra‐spacecraft transmissions of sensor and SpaceWire data” published by the International Journal of Satellite Communications and Networking [1]. Nonetheless, the key elements that outline the novelty of this paper and in the same time distinguish it from the previous work described in [1] derive from: (1) the use of validation test scenarios defined by of Eu:CROPIS mission system requirements, (2) the use of proprietary, CDS designed, TRL6 WIU Engineering Qualification Models (EQM), which exhibit both a custom hardware design (compact solution for CDHU/PDHU implementation) and a custom software design (proprietary software running on an MCU), and (3) the use of SpW-capable equipment for WIUs tests and validation. Through the three major elements mentioned above, the work presented in this manuscript displays its benefits as compared to the previous work described in [1], [7], and [9].

[Comment #3] Clear the experimental results in the abstract and conclusion.

[Reply]: The Abstract was updated to clear the experimental results as follows (see lines 24-31):

The validation test scenarios of the WIUs are carried out with the use of STAR-Dundee SpW-capable equipment. The test results demonstrate the reliability of the deployed SpW-to-UWB WIUs devices for UWB wireless communications carried out within a space-shuttle. The SpW data was successfully transmitted across the intra-spacecraft wireless network in all experimental tests. The technology can be considered at maturity level TRL6 (functionality demonstrated in relevant environment) for LEO missions.

The conclusion was updated to clear the experimental results as follows (lines 600-612):

To conclude, the WIUs did successfully perform all functional tests as defined in accordance with the WiSAT-3, European Space Agency (ESA)-funded project, namely Eu:CROPIS mission system requirements. The SpW data was successfully transmitted across the intra-spacecraft wireless network in all experimental tests. As it was expected, the adopted architecture proved to be viable and functional. The technology can be considered at maturity level TRL6 (functionality demonstrated in relevant environment) for LEO missions.

The suggested approach can be taken into consideration for future satellite missions, as well, even though it is based on the needs of the Eu:CROPIS mission. SpW, the top communication system for both present and future spacecraft, is envisioned by this approach. The solution also uses ISA100 over IEEE 802.15.4 UWB PHY, which have shown to be the most dependable wireless intra-satellite communication technologies among those that were studied by CCSDS [3].   

[Comment #4] Improve the quality of Fig 5.

[Reply]:  The quality of the figure has been improved, as required. An updated version of Figure 5 has been inserted into the article.
